# BML: A High-performance, Low-cost Gradient Synchronization Algorithm for DML Training

**Songtao Wang**[1,2]**, Dan Li**[1]**, Yang Cheng**[1]**, Jinkun Geng**[1]**,**
**Yanshu Wang**[1]**, Shuai Wang**[1]**, Shutao Xia**[1,2] **and Jianping Wu**[1]

[1]Department of Computer Science and Technology, Tsinghua University
[2]Graduate School at Shenzhen, Tsinghua University

## Abstract

In distributed machine learning (DML), the network performance between machines significantly impacts the speed of iterative training. In this paper we propose *BML*, a new gradient synchronization algorithm with higher network performance and lower network cost than the current practice. *BML* runs on BCube network, instead of using the traditional Fat-Tree topology. *BML* algorithm is designed in such a way that, compared to the parameter server (PS) algorithm on a Fat-Tree network connecting the same number of server machines, *BML* achieves theoretically $\frac{1}{k}$ of the gradient synchronization time, with $\frac{k}{5}$ of switches (the typical number of $k$ is 2~4). Experiments of LeNet-5 and VGG-19 benchmarks on a testbed with 9 dual-GPU servers show that, BML reduces the job completion time of DML training by up to 56.4%.

## 1  Introduction

Machine learning (ML) has become a core service in large companies [14]. The scale of modern ML training can be huge [17, 7, 4]. From our survey of a large internet company, a CTR (click through rate) estimation task trains a model of >100 billion features with >1PB training data. Given the memory size and processing capability of today's commodity machines, it is inevitable to run distributed machine learning (DML) on multiple machines [10, 18]. For instance, the internet company under survey currently uses several hundreds of dedicated machines to carry out the training for CTR estimation. With the ever-increasing training data and model sizes, it is expected that even larger-scale DML will appear in the near future.

A typical ML training task trains a model iteratively until the parameters converge. In the widely-used *gradient descent* optimization method, in each iteration the algorithm uses a *minibatch* of training data to compute a *gradient*, which decides the changes to make to the parameters trained by the previous iteration. In DML, every machine iteratively trains a *sub-minibatch* of data and synchronizes the gradients with other machines. Ideally, more machines help reduce the training time. However, it has been shown that, when more machines are used in DML, we have to set a *smaller sub-minibatch size* per machine, so as to keep the aggregated minibatch over all the machines with a reasonable size. Otherwise, the large aggregated minibatch may cause the training to quickly converge to a worse model. For instance, a recent work from Facebook discloses that their translation service cannot currently train on large minibatches without degrading model quality [14].

A side effect of smaller sub-minibatch size per machine in larger-scale DML is the break of computation/communication balance. For example, an experiment from Amazon shows that [23], if setting the batch size on a GPU as 16, the processing time per batch stays stable from 1 GPU to 128 GPUs; while if setting the batch size on a GPU as 2, the processing time per batch under 128 GPUs increases by more than 6 times compared with the time per batch under a single GPU, because of the

dominating communication cost. Therefore, in order to run DML in large scale, we need to carefully design the network with minimized synchronization overhead among machines.

The widely-used network topology to run DML in today's data centers is Clos network, or Fat-Tree [5]. Although Fat-Tree achieves great success in providing uniform network performance to cloud computing applications, it may not well match the traffic model of gradient synchronization in DML. Running the typical parameter server (PS) synchronization algorithm in Fat-Tree, each synchronization flow needs to traverse multiple hops of switches before being aggregated. It not only hurts the gradient synchronization performance, but also wastes the bandwidth/link resource.

In this paper, we suggest using BCube [12] as the underlying network topology for DML training, and design a novel distributed gradient synchronization algorithm on top of BCube, called *BML*. BCube is a recursive network topology composed of commodity switches and servers with $k$ (the typical value of $k$ is 2∼4) interfaces. The synchronization algorithm of *BML* is designed in such a way that, compared to the PS algorithm running a Fat-Tree network connecting the same number of server machines, *BML* running on a BCube network can theoretically achieve $\frac{1}{k}$ of the gradient synchronization time, with only $\frac{k}{5}$ of switches.

We have implemented *BML* in TensorFlow. We run two representative public deep learning benchmarks, namely, LeNet-5 [15] and VGG-19 [22], on a testbed with 9 dual-GPU servers. The experiment results show that, *BML* can reduce the job completion time of DML training by up to 56.4% compared with the PS algorithm on Fat-Tree network. The advantage of *BML* is higher when the sub-minibatch size per machine is smaller, which is important for large-scale DML to guarantee the model accuracy.

## 2  Background and Motivation

**DML Models and Notations:** DML can run on multiple CPUs/GPUs in a machine or on multiple machines. In this work we focus on *DML network among machines*. In order to decouple the inter-machine and intra-machine communications, throughout this paper we simply take one machine as a single training worker, though the machine can be equipped with multiple GPUs.

Based on splitting whether the training data or the model parameters onto multiple machines, DML can be divided into data-parallel and model-parallel ones. In data-parallel DML, each machine uses a shard of training data to compute the gradients; while in model-parallel DML, a machine computes gradients for part of the model. In this work we focus on *data-parallel DML*. In each iteration, every machine trains local gradients for the entire model based on its sub-minibatch of training data, and synchronizes the gradients with other machines. The aggregated gradients are calculated upon all the machines' local gradients, which are then applied to the model update.

According to the tradeoff between gradient freshness and computing resource utilization, there are three typical synchronization modes. 1) Bulk synchronous parallel (BSP). 2) Total asynchronous parallel (TAP). 3) Stale synchronous parallel (SSP). Given a predefined accuracy of the trained model, it is difficult to tell which synchronization mode runs the fastest in practice. BSP wastes the computation resource of some faster machines, but fully follows the sequential behavior as trained by a single machine. TAP makes full utilization of the computing resource, but the convergence speed is unpredictable with the possibility of no convergence at all [10]. SSP lies between the two with proven convergence [9, 26]. In this work we focus on *BSP synchronization*, which is widely used in modern ML applications [11, 24].

Table 1: Notations used throughout the paper

| Notation | Meaning |
|---|---|
| $N$ | The total number of servers in a DML network |
| $P$ | The size of full gradients |
| $T_F$ | The theoretical time to transmit the full gradients by full link speed |
| $T_C$ | The theoretical time to transmit a gradient piece by full link speed |

We summarize the notations used throughout the paper in Table 1. $N$ denotes the total number of server machines in a DML network, $P$ denotes the size of full gradients for the trained model, and $T_F$ denotes the theoretical time to transmit the full gradients by full link speed. Many gradient

synchronization algorithms divide the full gradients into multiple pieces during synchronization, and we use $T_C$ to denote the theoretical time to transmit a gradient piece by full link speed.

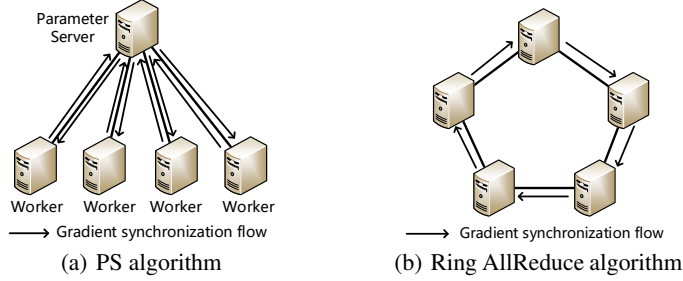

Figure 1: Gradient synchronization algorithms.

**Gradient Synchronization Algorithm:** There are two representative algorithms to synchronize the gradients among machines, namely, PS algorithm [17] and Ring AllReduce algorithm [19]. The PS algorithm is shown in Fig. 1(a), in which a logical parameter server (composed of a number of physical servers) interacts with every worker for parameter update. Each worker pushes its local gradients to the parameter server after training in an iteration, and pulls the aggregated gradients from the parameter server before going to the next iteration. With more workers in DML, the amount of traffic exchanged with the parameter server also increases.

The Ring AllReduce algorithm is widely used in HPC, as shown by Fig. 1(b). If we run the Ring AllReduce algorithm for gradient synchronization, all the machines are organized as a logical ring and the algorithm includes two stages. In the scatter-allreduce stage, it takes $N - 1$ steps for each machine to aggregate $\frac{1}{N}$ of the gradients; in the allgather stage, it takes $N - 1$ more steps for each machine to get a complete set of the updated gradients. Each step takes the time of $\frac{1}{N} * T_F$, if full link speed can be used by every machine. The theoretical gradient synchronization time(GST) in the Ring AllReduce algorithm is thus $\frac{2*(N-1)}{N} * T_F$.

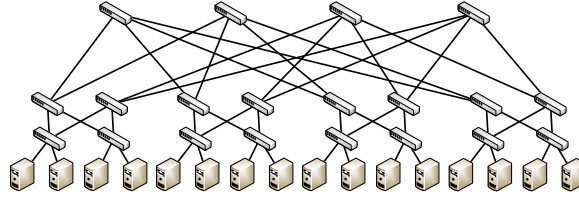

Figure 2: A Fat-Tree network with 16 servers.

**Physical Network Topology:** Fat-Tree is the current practice of physical network topology in most commercial data centers, as shown in Fig. 2. When running the PS algorithm for gradient synchronization in Fat-Tree, it is flexible to place the parameter servers and workers. One way is to partition the machines into two clusters, one for parameter servers and the other for workers. Another way is to implement the logical parameter server in a *P2P* manner, i.e., every machine plays as both a worker and a parameter server. For reducing the gradient synchronization time, the latter way makes better utilization of the network bandwidth. In every iteration, each machine is responsible for aggregating $\frac{1}{N}$ of the gradients and broadcasting to all the other machines. Hence, during gradient synchronization each machine pushes $\frac{P}{N}$ local gradients to, and pulls $\frac{P}{N}$ aggregated gradients from, every other machine. Since the Fat-Tree network provides non-blocking bandwidth, the theoretical GST for P2P based PS algorithm in Fat-Tree is $\frac{2*(N-1)}{N} * T_F$.

If running the Ring AllReduce algorithm in a Fat-Tree network, the theoretical GST is also $\frac{2*(N-1)}{N} * T_F$, by utilizing the bidirectional bandwidth of every server link. Since the two gradient synchronizatioin algorithms achieve the same GST in a Fat-Tree network and the PS algorithm is more widely implemented in modern DML frameworks [4, 1, 6], in this paper we only take the PS algorithm in Fat-Tree as the benchmark.

**Motivation of *BML* Design:** Although Fat-Tree achieves great success in providing uniform network performance to cloud computing applications, in this paper we argue that Fat-Tree does not well match the traffic pattern of DML training. When running the PS algorithm, each synchronization flow needs to traverse multiple hops of switches before being aggregated, which not only hurts

the gradient synchronization performance, but also wastes the bandwidth/link resource. We seek to design a new gradient synchronization algorithm on alternative network topology, which can achieve less GST with lower hardware cost. The new network topology and synchronization algorithm atop can be used to build a server cluster purposely for DML training.

## 3 *BML* Design

### 3.1 BCube Topology

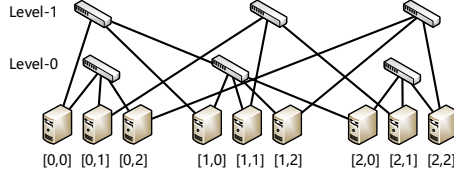

Figure 3: The topology of BCube(3,2).

We select BCube [12, 8] as the underlying physical network topology for DML training. BCube($n$,$k$) is a recursive topology composed of commodity servers with $k$ (the typical value of $k$ is 2∼4) network interfaces and switches with $n$ ports. Fig. 3 shows an example of BCube(3,2) topology. Note that modern commodity GPU servers used in ML training have multiple PCIe slots. Besides installing the GPU cards, it is easy to equip $k$ network interfaces on a GPU server. Since in this paper we assume network is the bottleneck rather than computation power, plugging in several NICs rather than GPUs into PCIe slots are reasonable. BCube switches are organized in $k$ levels (identified from 0 to $k - 1$). Each server uses one interface to connect a switch from one level. A BCube server can be denoted by an ID of $s = [v_{k-1}..., v_1, v_0](v_i \in [0, n - 1], \forall i \in [0, k - 1])$. The links connecting level-$i$ switches are called level-$i$ links. It is worth noting that BCube switches are not directly connected with each other. If the shortest path between two BCube servers traverses $q$ switches, the two servers are called *q-hop neighboring servers*. For instance, in Fig. 3 each server has four 1-hop neighboring servers.

A BCube($n$,$k$) network contains $n^k$ servers and $k * n^{k-1}$ switches. For instance, a BCube(16,4) network has 65536 servers. Hence, BCube can extend to large scale on commodity servers and switches. Compared with the classical 3-layer Fat-Tree topology [5], BCube pays the cost of using more network interfaces on servers. However, the number of switches required in BCube is much less than that in Fat-Tree. Considering the situation that many servers in modern data centers are equipped with at least dual ports [16], the cost of BCube can be lessened. In order to connect a total number of $N$ servers by $n$-port switches, a BCube network needs $\frac{k*N}{n}$ switches, while a Fat-Tree network needs $\frac{5*N}{n}$ switches. Given the typical value of $k$ in BCube is $2 \sim 4$, the number of switches required in BCube is 40∼80% of that in Fat-Tree. Since the cost of a server NIC is much less than that of a switch, the total network cost in a BCube network is considerably less than that in a Fat-Tree network connecting the same number of servers.

### 3.2 Gradient Synchronization Algorithm

In every training iteration, servers take a fully-distributed way to synchronize the gradients. As illustrated in Table 1, we use $N$ to denote the total number of servers in a BCube($n, k$) network, with $N = n^k$. $k$ gradient synchronization threads (each thread identified by $e \in [0, k - 1]$) are run simultaneously on each server. The full set of gradients, with the size of $P$, are equally split into $k * N$ gradient pieces. The theoretical time to transmit a gradient piece by full link speed is thus $T_C = \frac{T_F}{k*N}$. Each synchronization thread $e$ on a server $q$ is responsible for aggregating one gradient piece, and thus we can use $g =< e, q >$ to identify a gradient piece. We further use $g(S)$ to denote the gradient piece $g$ that is aggregated on the set of servers $S$. Obviously, the initial state of a gradient piece $g$ on a server $s$ is $g(s)$, and the final state of the gradient piece after synchronization is $g([*, *, ..., *])$.

Each gradient synchronization thread on a server runs the algorithm in two stages, namely, aggregation stage and broadcast stage. In the aggregation stage, the thread exchanges gradient pieces with the same thread on other servers, and aggregates one gradient piece. In the broadcast stage, the

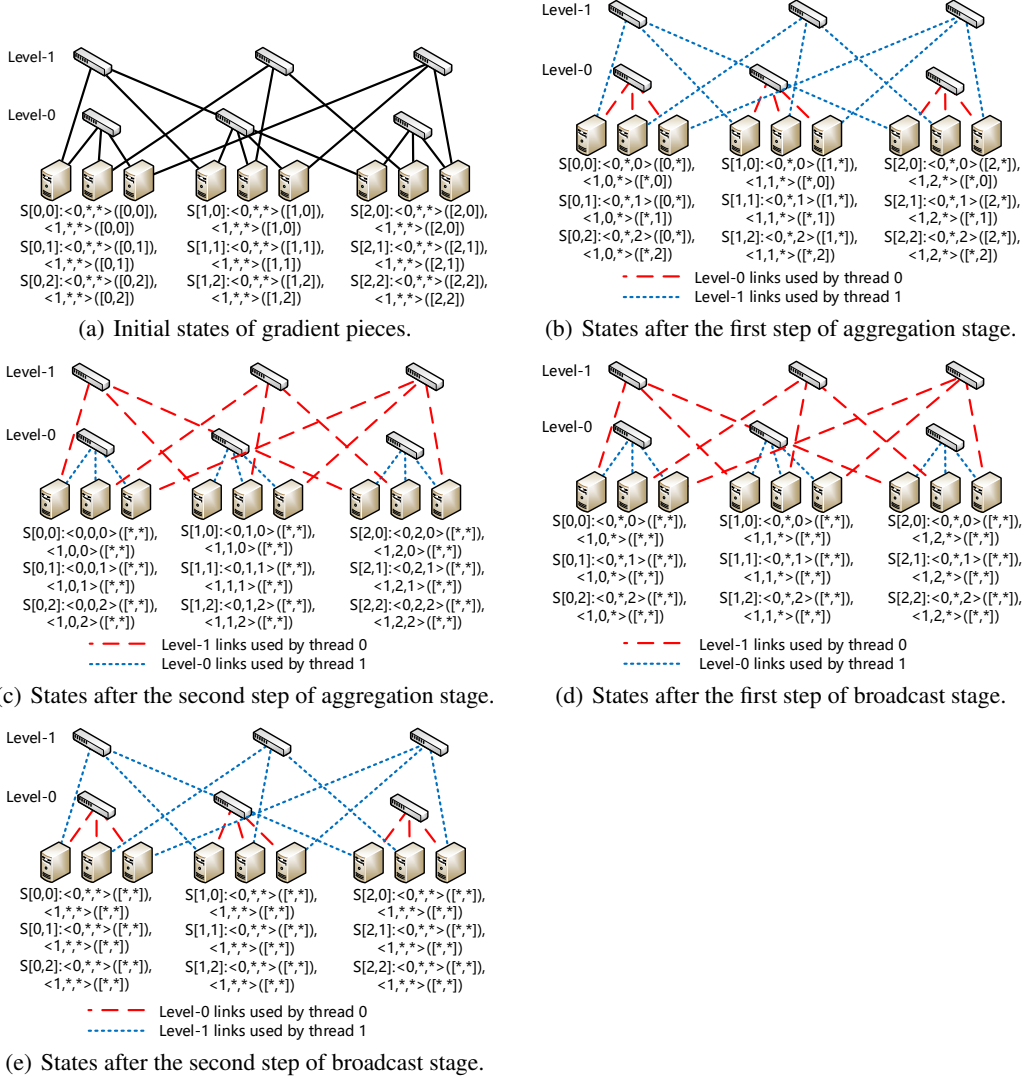

(a) Initial states of gradient pieces.

(b) States after the first step of aggregation stage.

(c) States after the second step of aggregation stage.

(d) States after the first step of broadcast stage.

(e) States after the second step of broadcast stage.

Figure 4: *BML* Gradient synchronization algorithms.

thread broadcasts its aggregated gradient piece to all the other servers. Finally, every server gets a complete set of aggregated gradients, which are used to update the parameters. Both aggregation stage and broadcast stage take $k$ steps in each. In what follows we first use an example in BCube(3,2) to demonstrate the process of *BML* algorithm. After that we describe the generalized algorithm.

**An Example of *BML* Algorithm in BCube(3,2):** As shown by Fig. 4(a), in a BCube(3,2) network every server runs two gradient synchronization threads. At the beginning each server $s$ only has the local gradients trained by itself. These gradients are split into 18 (=9*2) pieces identified from $< 0, 0, 0 > (s)$ to $< 1, 2, 2 > (s)$. The theoretical time to transmit a gradient piece is $T_C = \frac{T_F}{18}$.

In the first step of the aggregation stage, every server exchanges gradient pieces with its 1-hop neighboring servers, as shown in Fig.4(b). Take the gradient synchronization thread 0 on server $[0, 0]$ as an example. It sends 3 local gradient pieces $< 0, *, 1 > ([0, 0])$ to server $[0, 1]$ and 3 pieces of $< 0, *, 2 > ([0, 0])$ to server $[0, 2]$, respectively. At the same time, it also receives 3 gradient pieces of $< 0, *, 0 > ([0, 1])$ from server $[0, 1]$ and 3 pieces of $< 0, *, 0 > ([0, 2])$ from server $[0, 2]$. Together with the local gradient pieces of $< 0, *, 0 > ([0, 0])$, it aggregates 3 gradient pieces $< 0, *, 0 >$ based on the 3 servers under the same level-0 switch. The *partially-aggregated* result is $< 0, *, 0 > ([0, *])$. Each gradient synchronization thread on every server makes similar partial aggregation. Note that in this step synchronization thread 0 on all servers use level-0 links only while thread 1 on all servers use level-1 links only. Therefore, it takes the theoretical time of $6 * T_C$ to complete this step.

In the second step, every synchronization thread further exchanges its partially-aggregated gradient pieces with 1-hop neighboring servers in another level, i.e., thread 0 taking level-1 links and thread 1 taking level-0 links. Fig. 4(c) shows the process. We again use synchronization thread 0 on server $[0, 0]$ as the example. It sends the partially-aggregated gradient piece $< 0, 1, 0 >$ ($[0, *]$) to server $[1, 0]$ and $< 0, 2, 0 >$ ($[0, *]$) to server $[2, 0]$, respectively. At the same time it receives partially-aggregated gradient pieces $< 0, 0, 0 >$ ($[1, *]$) from server $[1, 0]$ and $< 0, 0, 0 >$ ($[2, *]$) from server $[2, 0]$, respectively. Together with the local partially-aggregated piece $< 0, 0, 0 >$ ($[0, *]$), the fully-aggregated result for gradient piece $< 0, 0, 0 >$ based on all the 9 servers is made, represented as $< 0, 0, 0 >$ ($[*, *]$). Similarly, each synchronization thread makes full aggregation of one gradient piece. This step takes the theoretical time of $2 * T_C$, and ends the aggregation stage.

In the first step of the broadcast stage, every thread on a server broadcasts its fully-aggregated gradient piece to 1-hop neighboring servers, as shown in Fig.4(d). Thread 0 takes level-1 links and thread 1 takes level-0 links. We still use thread 0 on server $[0, 0]$ as the example. It broadcasts the gradient piece of $< 0, 0, 0 >$ ($[*, *]$) to servers $< 1, 0 >$ and $< 2, 0 >$ simultaneously. At the same time, it receives gradient piece $< 0, 1, 0 >$ ($[*, *]$) from server $< 1, 0 >$ and receives piece $< 0, 2, 0 >$ ($[*, *]$) from server $< 2, 0 >$. This step takes the theoretical time of $2 * T_C$. After this stage, each thread on a server gets 3 fully-aggregated gradient pieces.

In the second step, every thread broadcasts its 3 fully-aggregated gradient pieces to 1-hop neighboring servers in another level, namely, thread 0 taking level-0 links while thread 1 taking level-1 links. It is easy to infer that this step takes the theoretical time of $6 * T_C$, and the final state is shown in Fig. 4(e). This is the end of the broadcast stage. Each server gets a complete set of 18 fully-aggregated gradient pieces, which is used to update the model parameters. The gradient synchronization traffic makes full utilization of the network bandwidth, and the total theoretical GST for a Bcube(3,2) network is $16 * T_C = \frac{8}{9} * T_F$.

**General Algorithm:** Next we describe the general *BML* algorithm run in thread $t$ of server $a$ in a BCube($n$,$k$) network. Note that $k$ threads simultaneously run the same algorithm on each server. As aforementioned, the algorithm takes $k$ steps in the aggregation stage and $k$ steps in the broadcast stage. In the aggregation stage, the $k$ steps use the level-($t\ mod\ k$), level-($(t+1)\ mod\ k$), ..., level-($(t+k-1)\ mod\ k$) links respectively, to synchronize the gradient pieces with 1-hop neighboring servers. Therefore, in every step, the links taken by the $k$ gradient synchronization threads on server $a$ do not collide with each other. In step $w$ ($w \in [0, k-1]$), thread $t$ sends $\frac{N}{n^{(w+1)}}$ gradient pieces to, and receives the same number of gradient pieces from, each of its 1-hop neighboring servers. The ID's of the exchanged gradient pieces are specified by the functions of CalcGset() and CheckDigits(). Hence, step $w$ takes the time of $\frac{N}{n^{(w+1)}} * (n-1) * T_C$. Taking all the $k$ steps together, the total time in the aggregation stage is $(N-1) * T_C$. The broadcast stage works similarly with the aggregation stage, except that in each step a thread broadcasts the fully-aggregated gradient pieces to its 1-hop neighboring servers instead of making aggregation. It takes the same time as the aggregation stage. The total GST for a BCube($n$,$k$) network is thus $2 * (N-1) * T_C = \frac{2*(N-1)}{k*N} * T_F$.

Therefore, compared with the theoretical GST of $\frac{2*(N-1)}{N} * T_F$ when running PS algorithm on a Fat-Tree network, *BML* algorithm on a BCube($n$,$k$) network theoretically uses only $\frac{1}{k}$ of GST, with less network cost. One may argue that we can also equip multiple NICs on a server to connect a Fat-Tree network. However, it leads to more switches and several times higher network cost.

## 3.3 The Importance of both BCube Topology and BML Algorithm

We choose BCube as the underlay DML network to replace Fat-Tree for two reasons. First, as one of the server-centric network topologies, BCube has well-known lower network cost than Fat-Tree. Second, compared with other server-centric network topologies such as DCell [13] and FiConn [16], BCube has a nice topological feature that it can better speedup gradient synchronization in DML. This speedup is realized by the *BML* algorithm, which works in a hierarchical way and the intermediate results are aggregated to reduce the traffic load. On the contrary, if we run traditional PS algorithm in BCube, it will double the GST compared with running BML. It is because the PS algorithm works in a flat way, which not only occupies more links for an individual flow but also fails to make aggregation during synchronization.

---

**Algorithm 1**

---

**Gradient synchronization algorithm of thread $t$ on server $a$ in a BCube($n$,$k$) network**

$s.G$: A set of gradient pieces $G$ on a server $s$

$a.GF$: The full set of gradient pieces on server $a$

$N_l(a)$:The set of server $a$'s 1-hop neighboring servers under the same level-$l$ switch

$s.d[i]$: The $i$-th digit of a server $s$' ID $[v_{k-1}, ..., v_1, v_0]$

$g.d[i]$: The $i$-th digit of a gradient piece $g$'s ID $< e, v_{k-1}, ..., v_1, v_0 >$

  1: $a.GF \leftarrow$ full gradient pieces on server $a$ by local training

  2: **for** $w \in [0, k-1]$ **do**

  3:     RunAggregation($w$)

  4: **for** $w \in [0, k-1]$ **do**

  5:     RunBroadcast($w$)

**function** RunAggregation($w$)

  6: $l \leftarrow (w+t)mod(k)$

  7: **for** each server $s \in N_l(a)$ **do**

  8:     $a.G \leftarrow$ CalcGset($s$,$w$,$a.GF$)

  9:     Transmit $a.G$ to $s$ and receive $s.G$ from $s$ through level-$l$ link

10: $a.GF \leftarrow$ updated full gradient set by aggregating $a$'s local gradient pieces and received gradient pieces from $N_l(a)$

**function** RunBroadcast($w$)

11: $l \leftarrow (k-1-w+t)mod(k)$

12: $a.G \leftarrow$ CalcGset($a$,$k-1-w$,$a.GF$)

13: **for** each server $s \in N_l(a)$ **do**

14:     Transmit $a.G$ to $s$ and receive $s.G$ from $s$ through level-$l$ link

15: $a.GF \leftarrow$ updated full gradient set by replacing $a$'s local gradient pieces with received gradients pieces from $N_l(a)$

**function** CalcGset($s$,$w$,$a.GF$)

16: $R \leftarrow \emptyset$

17: **for** each gradient piece $g \in a.GF$ **do**

18:     **if** CheckDigits($g$,$s$,$w$)=True **then**

19:        put $g$ into $R$

20: **return** $R$

**function** CheckDigits($g$,$s$,$w$)

21: **for** $i \in [0, w]$ **do**

22:     $j \leftarrow (i+t)mod(k)$

23:     **if** $g.d[j] \neq s.d[j]$ or $g.d[k] \neq t$ **then**

24:        **return** False

25: **return** True

---

On the other side, we can run BML-like hierarchical synchronization algorithm on a Fat-Tree network, where servers are first grouped by edge switches, then grouped by aggregation switches, and finally grouped by pods. But this synchronization algorithm results in the same GST as running PS algorithm in Fat-Tree. The main reason is that the single NIC at each server in Fat-Tree cannot speedup the synchronization. Although there are more switches in Fat-Tree than in BCube, they are amortized since each synchronization flow in Fat-Tree traverses multiple switches. If we use $k$ NICs at each server to connect the Fat-Tree network, as BCube server does, it can indeed lead to the same GST as BCube; but we need $k$ times more switches to support the same number of servers, which further increases the cost gap between BCube and Fat-Tree.

We use the Fig.5 to demonstrate the GSTs when running PS-based algorithm and BML (hierarchical synchronization algorithm) on both Fat-Tree and BCube networks. It clearly demonstrates the necessity to choose both BCube topology and the BML algorithm for DML.

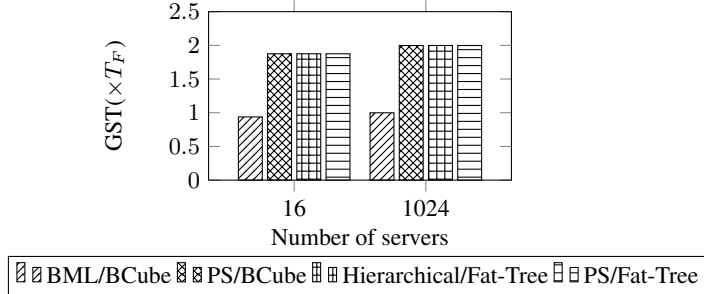

Figure 5: GST of running BML (hierarchical synchronization) and PS-based algorithms on BCube and Fat-Tree networks respectively. 16 servers refer to BCube(4,2) and Fat-Tree(4), while 1024 servers refer to BCube(32,2) and Fat-Tree(16).

## 4    Implementation and Experiments

### 4.1    Implementation

We implement *BML* in TensorFlow [4]. The current gradient synchronization algorithms implemented in the open-source version of TensorFlow is the PS algorithm. Our implementation of *BML* includes 4550 lines of C++ code and 702 lines of Python code. It contains three main modules, namely, *sending module*, *receiving module* and *management module*. The sending module gets the gradient pieces from the *sending queues* (enqueued by the management module) and sends them to the corresponding neighbouring servers. The receiving module receives the gradient pieces from neighbouring servers and submits them to the management module. The management module bridges the other two modules, maintains the sending queues, and aggregates the gradient pieces based on the remote ones and local ones in the aggregation stage.

It is worth noting that, for deep learning, the neural network model has more than one layers. In the back-propagation algorithm [20, 21], the gradients of the model are computed layer by layer. When the gradients for one layer are computed out, they can be transmitted while the gradients for other layers are still under computation. The time of computing and transmission can thus overlap. Therefore, in our implementation we divide the gradients for each layer of the model into $k * N$ pieces in a BCube($n$,$k$) network (with $N = n^k$), instead of simply dividing the gradients for the entire model into $k * N$ pieces. In this way, the gradient synchronization load on each server is well balanced.

### 4.2    Experiment Setting

We build a BCube(3,2) testbed with 9 dual-GPU servers and multiple 40GE switches. Each server is equipped with two Nvidia Tesla K40C GPUs, two Intel Xeon E5 CPUs, 64GB DRAM and two 40Gbps Mellanox Connectx-3 NICs. To compare *BML* with Fat-Tree, we also build a Fat-Tree network with the same number of GPU servers. Since the network size is not very large, we simply use a single 40GE switch to connect all the 9 servers, with each server using one NIC to connect the switch. It fully *emulates* the Fat-Tree network, as the network bandwidth is non-blocking. We run the P2P based PS algorithm for gradient synchronization in Fat-Tree, where each server plays as both a parameter server and a worker. RoCE (RDMA over Converged Ethernet) is used as the transport protocol in all these networks.

We run two representative public deep learning benchmarks, namely, LeNet-5 [15] and VGG-19 [22], in each network. LeNet-5 has a 5-layer model , with a total number of 3.27 million parameters. The size of full gradient is about 12.5MB (with single-precision floating point). The model of VGG-19 contains 19 layers and the total number of parameters is about 143.65 million. The full gradient size is thus 548MB. MNIST [3] and ImageNet [2] datasets are used as training data for LeNet-5 and VGG-19 respectively. To study the impact of minibatch size, we set different sub-minibatch sizes on a training server in different rounds of experiments. Since we only focus on the training speed of DML, we fix the number of iterations trained in each round of the experiment as 1000 and measure the *job completion time(JCT)* of the benchmark.

## 4.3 Results and Analysis

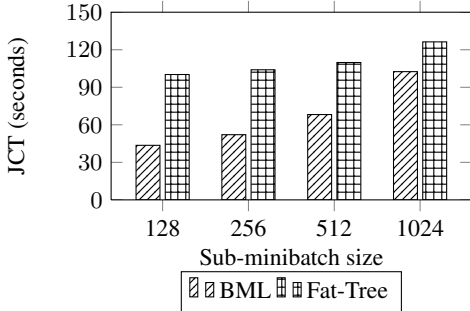

Figure 6: Experiment result of LeNet-5.

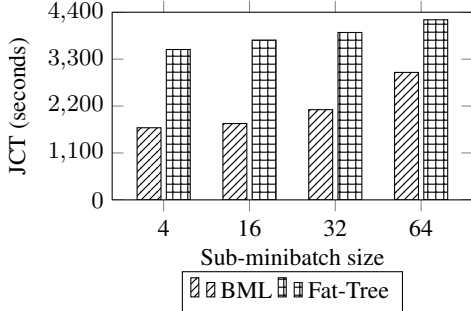

Figure 7: Experiment result of VGG-19.

**LeNet-5:** Fig. 6 illustrates the results for LeNet-5 benchmark on MNIST dataset. We set the sub-minibatch size on each server as 128, 256, 512 and 1024 in different rounds. Compared with Fat-Tree, *BML* reduces the JCT by 18.7%~56.4%. The gain comes from the following two causes. First, the theoretical GST in a BCube($n,k$) is $\frac{1}{k}$ of that in Fat-Tree. With $k = 2$ in this experiment, the GST of each training iteration in *BML* should be about half of that in Fat-Tree. Considering the computation time, theoretically *BML* should reduce the JCT by 0%~50% compared with Fat-Tree. Second, the current implementation of the PS algorithm in TensorFlow maps the gradients to the parameter servers on per-tensor basis [25]. As different tensors have different sizes, the loads on the Fat-Tree servers are not balanced. Hence, in the experiments we find that in some cases *BML* can reduce the JCT by more than 50%.

We also observe that, with smaller sub-minibatch size on a server, the performance gap between *BML* and Fat-Tree is larger, because the communication cost has a higher weight in the whole training job. As introduced in Section 1, in order to scale DML to large size without degrading the model quality, usually we have to set a relatively small sub-minibatch size per server. The experiment demonstrates that *BML* has particular advantage in this scenario.

**VGG-19:** The results for VGG-19 benchmark on ImageNet dataset is shown in Fig. 7. The model size of VGG-19 is much larger than LeNet-5, so it takes more time than LeNet-5 to finish the 1000 iterations of training. However, the performance gap between the three DML networks are very similar with that in LeNet-5. Generally, *BML* reduces the JCT by 29.2%~52.1% compared with Fat-Tree network.

## 5 Conclusion

In this paper we design *BML*, a new DML gradient synchronization algorithm with higher performance and lower cost. *BML* runs on BCube topology instead of the commonly-used Fat-Tree network in current data centers. Compared with the PS algorithm running on a Fat-Tree network connecting the same number of servers, *BML* achieves $\frac{1}{k}$ of of the GST while using only $\frac{k}{5}$ switches. The experiments of typical deep learning benchmarks on Tensorflow also validate the performance gains of *BML*.

## 6 Acknowledgement

The work was supported by the National Key Basic Research Program of China (973 program) under Grant 2014CB347800, and the National Natural Science Foundation of China under Grant No. 61522205, No. 61772305, No. 61432002, No. 61771273. Dan Li is the corresponding author of this paper.

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
