[Reviews · NeurIPS 2018]

Reviewer 1



This paper designs, implements and analyzes a gradient aggregation algorithm for a network architecture following the BCube (Guo et al 2009) topology. The context of distributed stochastic gradient descent is well-explained and the comparison between Fat-Tree and BCube is well laid out. The context around the BCube topology is very sparsely cited. Since adopting the BCube topology involves some investment in hardware and configuration, it would be useful to the readers to have an idea of its value outside of the scope of SGD optimization. The original impetus behind the introduction of BCube was fault tolerance and some bibliography to that effect (e.g. de Souza Couto 2015) would help. The gradual explanation of the algorithm in section 3.2, including its example, is well-appreciated, even though the dataflow in Figure 4 is difficult to grasp from visual elements. The paper may gain from citing and comparison with _Butterfly Mixing: Accelerating Incremental-Update Algorithms on Clusters_ (Zhao et al 2013) which also has a 1/k over a Fat-Tree AllReduce, though this falls very slightly out of the bounds of the strict BSP focus of the authors. In any case, the state diagrams represented there offer a more easily readable representation of the gradient transmission algorithm than Fig.4 The experimentation section is very useful as a validation, and an open-source release would be very appreciated. The use of the very small MNIST dataset is somewhat disappointing (CIFAR-10 would be a great alternative). Overall a good paper. Minor points: - GST (gradient synchronization time I assumed) is never spelled out

Reviewer 2



Motivated by the latency in network performance between machines during iterative training, this work proposes a new gradient synchronization algorithm that runs on BCube network. The performance of the algorithm shows significant decrease in the synchronization time with fewer number of switches. The paper is written well. While the authors mention that common distributed ml algorithm use is PS on a Fat Tree network, their choice of BCube for BML is not well justified. In terms of the presentation of the paper, too much details on the background and motivation is provided whereas more focus on the details of the algorithm and the evaluation would have added value. The primary concern of this paper is evaluation. In comparison to its competitor PS algorithm [1], the evaluation seems weak and not well done. While the primary take away in the evaluation is that the BML algorithm performs better than Fat Tree network, it is unclear if the synchronization algorithm is more valuable or the choice of BCube. An study or discussion to understand the contributions of choices to the performance is necessary. Background and motivation section in the paper does not represent and discuss related work which is also missing from the results section. It is important to have clear description of related work which also discusses comparisons to the presented approach. A couple of missing related literature [2, 3]. [1] Scaling Distributed Machine Learning with the Parameter Server [2] Distributed Training Strategies for the Structured Perceptron [3] http://papers.nips.cc/paper/4687-large-scale-distributed-deep-networks.pdf #### After author's reponse #### While, the authors have responded well to my comments, I am unsure how they would include the content in the paper, given the page limits. Based on author's response and reading other reviews, I am inclined to give the paper a positive score.

Reviewer 3



This paper observes that by putting network cards into computers, some of the router functionality can be performed for free by each compute node. This potentially means less router equipment, and less transmission time for averaging data. This paper then shows how to you how to do it for real, giving details of the averaging algorithm for 'k' network cards per compute node. For k=2 and small minibatches, they actually see their expected factor of 2 speedup experimentally. The algorithm has some little tricks in it that make it a useful contribution. It is nice that this technique can in many cases be combined with other ways to reduce communication overhead. The technique is more something to consider when setting up a new compute cluster. This is really a networking paper, since the technique describes how to aggregate [sum] and distribute any vector amongst N compute nodes. The vector could be a gradient, a set of model parameters, or, well, anything else. On the other hand, they actually demonstrate experimentally that it can work in practice for a fairly common machine learning scenario. ### After author review ### Margin notes: line 95: GST - where defined? line 226: codes --> code Some of the statements in the rebuttal seemed more understandably stated than the more difficult level of presentation in the paper. Occasionally I had to remember previous statements and infer comparisons by myself. As noted by reviewer #2, the comparison figure and its discussion helps bring out the practical importance of their paper. I agree with reviewer #1 that Figure 4 and its discussion needs a very devoted reader. If I had to make room, I might consider moving Figure 4, lines 165--203 and the dense notation use for it (148--157) to a Supplementary Material. The body of the paper might contain a much simplified version of Fig. 4 and its discussion, with just the thread 0,1 link activity during the two steps each of aggregation and broadcast, and refer the reader to the supplementary material for details of precisely which gradient piece is chosen for transmission so as to lead to a final state where all servers know all pieces.